# Room-Temperature Synthesis of Carbon Nanochains via the Wurtz Reaction

**DOI:** 10.3390/nano15050407

**Published:** 2025-03-06

**Authors:** Juxiang Pu, Yongqing Gong, Menghao Yang, Mali Zhao

**Affiliations:** Interdisciplinary Materials Research Center, School of Materials Science and Engineering, Tongji University, Shanghai 201804, China; 2230661@tongji.edu.cn (J.P.); 2331571@tongji.edu.cn (Y.G.)

**Keywords:** Wurtz reaction, stereochemical configuration, carbon nanochains, room temperature

## Abstract

In the field of surface synthesis, various reactions driven by the catalytic effect of metal substrates, particularly the Ullmann reaction, have been thoroughly investigated. The Wurtz reaction facilitates the coupling of alkyl halides through the removal of halogen atoms with a low energy barrier on the surface; however, the preparation of novel carbon nanostructures via the Wurtz reaction has been scarcely reported. Here, we report the successful synthesis of ethyl-bridged binaphthyl molecular chains on Ag(111) at room temperature via the Wurtz reaction. However, this structure was not obtained through low-temperature deposition followed by annealing even above room temperature. High-resolution scanning tunneling microscopy combined with density functional theory calculations reveal that the rate-limiting step of C–C homocoupling exhibits a low-energy barrier, facilitating the room-temperature synthesis of carbon nanochain structures. Moreover, the stereochemical configuration of adsorbed molecules hinders the activation of the C–X (X = Br) bond away from the metal surface and, therefore, critically influences the reaction pathways and final products. This work advances the understanding of surface-mediated reactions involving precursor molecules with stereochemical structures. Moreover, it provides an optimized approach for synthesizing novel carbon nanostructures under mild conditions.

## 1. Introduction

Carbon nanomaterials, including carbon nanotubes, fullerenes, graphene, metalated carbyne ribbons, one-dimensional organometallic polyynes, and their derivatives, possess unique physicochemical properties and hold broad application prospects in electronic information technology, biomedicine, and energy storage and conversion [1,2,3,4].

On-surface synthesis has emerged as a powerful strategy for constructing novel carbon nanostructures that are often inaccessible via conventional solution synthesis. To date, classical chemical reactions including Ullmann coupling, the Glaser reaction, and dehydrogenation have been successfully realized and extensively explored on noble metal substrates under ultrahigh vacuum (UHV) conditions [5,6,7,8]. Through the rational design of precursor molecules, a wide variety of carbon-based nanostructures, such as graphene nanoribbons, conjugated polymers, and so on, have been prepared with atomic precision using a bottom-up strategy involving the surface-assisted dehalogenation/dehydrogenation of alkanes, alkenes, and arenes, followed by polymerization. Typically, precursor molecules require external energy input, such as heat or light, to activate the dehalogenation or dehydrogenation process [9,10,11]. Moreover, metal-organic intermediates are inevitably generated during the formation of covalently bonded final products. Consequently, further annealing is essential to remove the metal atoms and obtain pure carbon nanostructures, thereby increasing the energy consumption and complicating the overall reaction process [12,13,14,15,16].

The Wurtz reaction, involving the coupling of alkyl halides through dehalogenation, has been a well-established strategy in solution chemistry for the preparation of hydrocarbons and polymers [17,18]. Compared with typical on-surface synthesis strategies, such as Ullmann coupling, studies on the Wurtz reaction on surfaces with metals as catalysts are sparse. In 2016, the Wurtz reaction was realized on metal surfaces with low activation energy. Notably, the molecular dimer was formed without the formation of an intermediate C–M–C structure [19]. These features make the preparation of nanostructures via the Wurtz reaction more efficient [19,20,21,22,23]. In 2023, covalently linked wavy chains were synthesized via the Wurtz reaction followed by Ullmann coupling, highlighting the potential of preparing carbon nanostructures using the Wurtz reaction; however, due to the involvement of Ullmann coupling, higher temperatures are required to induce the formation of wavy chains [24].

In this work, a precursor molecule with a binaphthalene backbone and two bromotoluene functional groups at the bay sites was chosen. Ag(111) substrate was employed as the catalytic platform due to its balanced catalytic activity for C–X (X = Cl, Br, and I) bond activation and moderate radical migration rate, compared to Cu and Au substrates, enabling the precise control over the reaction dynamics [12,19,22,24,25,26]. By combining scanning tunneling microscopy (STM) and density functional theory (DFT) simulations, it was demonstrated that the Wurtz reaction could occur on the Ag(111) substrate at room temperature, yielding the ethyl-bridged binaphthyl molecular chains. However, low-temperature deposition followed by post-annealing at a temperature even slightly above room temperature primarily resulted in singly debrominated radicals and a minor fraction of dimeric structures (see Figure 1). This suggests that the stereochemical adsorption configurations of the molecules hinders the triggering of the C–Br bond distant from the Ag(111) surface. This work demonstrates the potential of using the Wurtz reaction to prepare one-dimensional carbon nanostructures under mild conditions, which could further complement the fabrication of hydrocarbons and other complex and novel nanostructures with unique properties.

## 2. Experimental Section

All the STM experiments were performed in a variable temperature “Aarhus-type” STM (SPECS, Berlin, Germany) [27,28], consisting of a device for molecular evaporation, a standard sample preparation chamber, and a main chamber for sample characterization. The experiment was conducted under UHV condition with a base pressure of 1 × 10⁻^10^ mbar. The Ag(111) single crystal was cleaned through several cycles of 1.5 keV ion bombardment for 15 min, and subsequent annealing at 800 K for 10 min. After thoroughly degassing, the (R)-2,2-bis(bromomethyl)-1,1-binaphthalene molecules (purchased from Adamas, Shanghai, China) were sublimated from quartz crucibles of molecular evaporator onto Ag(111) surface under various conditions. Subsequently, the sample was transferred to the scanner at a temperature range of 100~150 K. The STM images were obtained in a constant-current mode, using an electrochemically etched tungsten tip.

DFT calculations were executed via the Vienna Ab Initio Simulation Package (VASP) (version 6.4.2) [29,30]. The interaction between ions and electrons was described using the projector-augmented wave method [31]. The Perdew–Burke–Ernzerhof (PBE) generalized gradient approximation exchange-correlation functional was used [32], and van der Waals (vdW) interactions were included through the dispersion-corrected DFT-D2 approach developed by Grimme [33]. The atomic configurations were refined using the conjugate gradient algorithm within VASP until the geometry was optimized to ensure that the forces on all free atoms were lower than or equal to 0.03 eV/Å. The Tersoff–Hamann method was used to simulate STM images [34], and the local density of states (LDOS) serves to estimate the tunneling current. To identify the transition state, the climbing-image nudged elastic band (CI-NEB) technique was employed [35]. The reaction route was refined until the forces along the path dropped to 0.03 eV/Å or less. For the CI-NEB simulations, eight intermediate configurations were employed to identify the transition state, and the energy curves were accordingly identified.

## 3. Results and Discussion

In this study, we choose the organic molecule (R)-2,2-bis(bromomethyl)-1,1-binaphthalene (abbreviated as BBMBN), which has methyl bromide functional groups at the bay sites of the binaphthalene backbone. Initially, BBMBN molecules were deposited on a cold Ag(111) substrate held at 135 K via thermal sublimation. This leads to the creation of hinged self-assembled structures (Figure 1a) along three directions with an angle difference of 122.5 ± 11.3° (Appendix A), which are dominated by the sextet symmetry of the Ag(111) lattice. The molecular coverage is estimated to be 94%. From the zoomed STM image (Figure 1b), each molecule exhibits a bright protrusion surrounded by a dark oval cap, which can be assigned to the bending-up bromomethyl group and binaphthalene of the trans-BBMBN molecule. The stereochemical configuration of the trans-BBMBN molecule (S-isomer) with one bromomethyl group bending up and the other bending down was demonstrated based on the DFT-optimized configuration (Figure 1e). The corresponding simulated STM image (Figure 1f) is in good accordance with the experimental observation. In addition, when BBMBN molecules were deposited on Ag(111) at slightly higher temperatures (150~180 K) with an estimated coverage of 65%, the self-assembly was found to be a more uniform parallelogram-like structure, which is balanced by van der Waals intermolecular interactions (Figure 1c). In the enlarged STM image (Figure 1d), molecules still exhibit a trans-configuration (S-isomer) and maintain a stereochemical adsorption structure. This is in stark contrast to the cis-configuration (R-isomer) (see Appendix A), which adopts a flatter geometry and has an adsorption energy that is 0.22 eV lower than that of the trans-configuration.

After annealing the sample at 350 K for 10 min, molecular self-assembly exhibits staggered rows of structure (Figure 2a). In the enlarged STM image (Figure 2b), each molecule displays a bright protrusion with a dark tail, which can be assigned to the upward methyl bromide and the binaphthalene backbone of the radical intermediates BBMBN* with one bromine atom removal. From the DFT-optimized geometry and the simulated STM image (Figure 2c,d), we can deduce that the C–Br bond of the trans-BBMBN molecule adjacent to the Ag(111) surface was broken, while the other one away from the metallic surface remained intact. The superimposed model in Figure 2b displays that the bromomethyl groups in adjacent molecules display a head-to-head arrangement, indicating that the BBMBN* molecular self-assembly is dominated by interactions with substrate silver atoms, and is also stabilized by intermolecular halogen bond interaction. In another BBMBN* molecular domain (Appendix A), the trans-halogen bonding between adjacent molecules is explicitly visible [36,37].

Interestingly, in a small area, rod-like structures with a length twice that of single BBMBN* molecules can be observed in every other staggered row with a total yield of 35% (Figure 2e). These structures can be deduced as dimerized molecules, indicating that the interaction with silver atoms is insufficient to stabilize the BBMBN* molecules. The homocoupling reaction becomes feasible once the bromine atoms are removed from the bromomethyl groups of the BBMBN molecules. In the enlarged STM image (Figure 2f), the organometallic intermediates involving C–Ag–C bonds, typically found in Ullmann reactions, are not observed. This is consistent with the features in previously reported Wurtz reactions on metallic substrates [24]. The rows of molecular dimers are separated by BBMBN* molecular arrays and exhibit an alternating orientation due to steric hindrance. Nevertheless, the formation of molecular chain structures was not observed upon post-annealing up to 350 K, indicating that the activation of the C–Br bond distant from the Ag(111) surface was hindered.

To explore diverse reaction schemes in different experimental conditions, the BBMBN molecules were deposited on Ag(111) substrates held at room temperature (RT, ~300 K), yielding self-assembled structures with a molecular coverage of 65% (see Figure 3a). In the zoomed STM image (Figure 3b), the morphology of each molecular motif can be clearly distinguished from that of the intact BBMBN molecule (Figure 1) or the radical species with one bromine atom removal (Figure 2). To determine the molecular species, the geometry of the biradical species after the complete removal of bromine atoms from the BBMBN molecule was calculated (see Figure 3c). The corresponding simulated STM image (Figure 3d) shows a cloud-shaped structure, featuring two semicircular parts on the upper and lower sections, respectively. The semicircular protrusion can be attributed to the tilted benzene ring in the binaphthyl groups. The simulated STM image based on the DFT-optimized geometry is in good agreement with the molecular morphology observed through STM imaging (Figure 3b). The protrusion in the upper section of the experimental imaged molecules indicates a slight upward bending of one biphenyl group, which is likely due to the interactions between the radicals. The small spots in the white circles can be deduced as the removed bromine atoms. Therefore, we deduced that the bromine atoms are completely removed after molecular deposition at RT, giving rise to the formation of the biradical species.

Subsequently, the sample was annealed at RT for 12 h, yielding a well-ordered chain-like structure (see Figure 4a,b and Appendix A). These chain structures feature an inclined configuration containing two lobes and a bright protrusion in the middle, with a periodicity of 5.68 ± 0.05 Å (Figure 4c). This can be inferred as the C–C homocoupling product formed after the cleavage of C–Br bonds in both bromomethyl groups. The DFT-optimized model of the Wurtz reaction-induced chain-like structure exhibits a stereochemical configuration, in which the geometry of the ethyl groups with naphthalene attached at both ends is the most elevated, followed by the phenyl group in naphthalene connecting to the ethyl group (Figure 4e). The simulated STM image reveals a bright protrusion in the middle and slight depressions above and below with an inclined configuration in each unit cell (Figure 4d), agreeing well with the experimental morphology and periodicity. Therefore, we conclude that we have formed the carbon nanochains consisting of periodic dinaphthalenes connected by ethyl groups via the Wurtz reaction on Ag(111) at room temperature.

To elucidate the mechanism of the Wurtz reaction of the BBMBN molecule on Ag(111), systematic DFT calculations were performed to unravel the pathways, including debromination, radical diffusion, and C–C homocoupling, employing the nudged elastic band (NEB) approach. To simplify the stereochemical structure of the BBMBN molecule, 2-(bromomethyl) naphthalene was employed in this model, as shown in Figure 5a. The calculated energy barrier associated with carbon–bromine bond scission on Ag(111) was 0.14 eV, which is slightly higher than the previously reported debromination energy barrier of 0.107 eV for BMBP molecules on the Ag(110) surface. Notably, the detachment of bromine atoms from the methyl group occurs at 150–200 K [19]. The DFT calculations demonstrate that the C–Br bonds remain intact for molecular deposition at temperatures around 150–180 K. Upon annealing to 350 K, selective activation of the C–Br bonds adjacent to the Ag(111) surface is observed, which can be attributed to the unique stereochemical configuration. The calculated C–C coupling energy barrier is 1.49 eV, significantly higher than the debromination and radical diffusion barrier (0.12 eV), see Figure 5c, indicating that C–C coupling is the rate-limiting step for the Wurtz reaction. Moreover, previous studies indicate that debromination from the aryl group with an energy barrier of 1.3–1.5 eV occurs at room temperature, which is similar to the energy barrier of C–C homocoupling calculated in our study, suggesting that C–C coupling is feasible at room temperature [38]. However, the energy barrier for C–H activation is 2.64 eV (Appendix A), and thus, further C–H activation is not observed at room temperature.

We also tried to deposit BBMBN molecules on the Au(111) substrate held at room temperature. Nevertheless, the obtained STM image is rather fuzzy (see Appendix A), which might be caused by the diffusion of BBMBN molecules, owing to the low adsorption energy on the Au(111) surface.

## 4. Conclusions

In conclusion, we have investigated the Wurtz reaction on the Ag(111) substrate under various conditions through high-resolution UHV-STM imaging and DFT calculations. By introducing methyl bromide functional groups at the bay sites of the 1,1′-binaphthalene backbone, we achieved the complete activation of C(sp^3^)–Br and C–C homocoupling under room-temperature deposition, leading to the formation of ethyl-bridged binaphthyl molecular chains through subsequent annealing at room temperature. DFT calculations reveal the low energy barriers for C–Br bond cleavage and C–C homocoupling, thereby demonstrating the feasibility of the Wurtz reaction at room temperature. By contrast, low-temperature deposition followed by annealing slightly above room temperature resulted primarily in partial debromination and minor dimerization. This can be attributed to the limited catalytic effect of the metal substrate on the activation of the C–Br bond in the remote methyl functional groups of precursor molecules. Therefore, the stereochemical configuration of adsorbed molecules plays a crucial role in determining the reaction pathways. It is proposed that the synthesis of continuous carbon nanostructures using precursor molecules with a stereochemical structure may be facilitated by deposition on a high-temperature metal surface.

## Data Availability

The data that support the findings of this study are available in the Appendix A of this article.

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
