# Peer review of "Room-Temperature Synthesis of Carbon Nanochains via the Wurtz Reaction"

_nanomaterials, 2025, doi:10.3390/nano15050407_

Round 1

Reviewer 1 Report

Comments and Suggestions for Authors

The manuscript entitled ‘’Room-temperature synthesis of carbon nanochains via the 2 Wurtz reaction '' contains some interesting findings on the bottom-up synthesis of ethyl-bridged binaphthyl molecular chains via the Wurtz reaction on Ag(111). The outcomes of the synthesis were characterized by scanning tunnelling STM studies supported by density functional theory calculations. The results are interesting and deserve publication.  In the following, I will try to make concrete suggestions on how to improve this article.

All the steps of the experiments were detailed and clearly described.

I think that the manuscript can be accepted for publication after the authors address some points that are not clear to me.

The minor points are the following:

  1. Authors should add a comment on the choice of Ag(111) as a metal substrate for the bottom-up synthesis. Is there any affinity of Ag with Br? You mentioned a catalytic effect on the activation of C-Br. Please add a comment on this point.
  2. Have they tried a different metal substrate like Au(111)?
  3. What is the substrate coverage found after each deposition process? Figures 1, 2 and 3 only report high-resolution images of the assembly.
  4. Is there any influence of the substrate on the orientation of the molecules after deposition?
  5. I suggest performing additional experimental studies like an X-ray photoelectron spectroscopy characterization in the future to deepen these studies.

Reviewer 2 Report

Comments and Suggestions for Authors

This study reports the observation of carbon nanochains formed by evaporating BBMBM on Ag(111) and then curing at room temperature. The authors proposed a mechanism of C-Br bond cleavage followed by C-C homocoupling. DFT calculation was included to support the hypothesis. This work is interesting but is fairly preliminary.  Additional experiments and controls are necessary to support the conclusions.

  1. The formation of the chain product was proposed to be C-Br bond cleavage and C-C homocoupling. Experimentally, only STM results were provided. A key experiment would be XPS showing the presence of Br in the starting material, and the absence of Br in the product. TOF-SIMS is another characterization that can provide elemental analysis on the surface composition. These additional characterizations are necessary to confirm the reaction and product structure.
  2. The image of Ag(111) alone should be provided to show the surface morphology without binaphthyl molecule.
  3. The initially deposited BBMBM before annealing for 12 h should be imaged and results included for comparison with the annealed sample.
  4. The authors mentioned that they obtained “continuous carbon nanostructures”. The STM only shows ~10-15 nm frame. Further zoom-out images are necessary to show the continuality of carbon nanostructures.
  5. Comparison with other substrates, such as Au(111), Cu(111) etc, both experimental results and DFT calculations, will be important to elucidate the role of Ag(111).
  6. “In contrast to room-temperature deposition, low-temperature deposition followed by annealing above room temperature resulted primarily in partial debromination and minor dimerization. This can be attributed to the limited catalytic effect of the metal substrate on the activation of the C-Br bond in the remote methyl functional groups of precursor molecules.” Additional characterizations such as XPS and TOF-SIMS mentioned in (1) and other substrates mentioned in (5) will be necessary to support this conclusion.
  7. Why is the R isomer of BBMBM necessary? Comparison with the (S) isomer would be informative.
  8. Scheme 1: define LT

Round 2

Reviewer 2 Report

Comments and Suggestions for Authors

The responses from the authors are adequate. I have no further questions.